# Multimodal detection of hateful memes by applying a vision-language pre-training model

**Yuyang Chen[1], Feng Pan[2]***

**1** Putnam Science Academy, Putnam, CT, United States of America, **2** Department of Radiology, Union Hospital, Tongji Medical College, Huazhong University of Science and Technology, Wuhan, China

\* uh_fengpan@hust.edu.cn

## Abstract

Detrimental to individuals and society, online hateful messages have recently become a major social issue. Among them, one new type of hateful message, "hateful meme", has emerged and brought difficulties in traditional deep learning-based detection. Because hateful memes were formatted with both text captions and images to express users' intents, they cannot be accurately identified by singularly analyzing embedded text captions or images. In order to effectively detect a hateful meme, the algorithm must possess strong vision and language fusion capability. In this study, we move closer to this goal by feeding a triplet by stacking the visual features, object tags, and text features of memes generated by the object detection model named Visual features in Vision-Language (VinVI) and the optical character recognition (OCR) technology into a Transformer-based Vision-Language Pre-Training Model (VL-PTM) OSCAR+ to perform the cross-modal learning of memes. After fine-tuning and connecting to a random forest (RF) classifier, our model (OSCAR+RF) achieved an average accuracy and AUROC of 0.684 and 0.768, respectively, on the hateful meme detection task in a public test set, which was higher than the other eleven (11) published baselines. In conclusion, this study has demonstrated that VL-PTMs with the addition of anchor points can improve the performance of deep learning-based detection of hateful memes by involving a more substantial alignment between the text caption and visual information.

## Introduction

Hateful messages, more commonly known as hate speech, have unfortunately almost become a ubiquitous phenomenon on social media. After reviewing the definitions of hateful messages from authoritative sources, such as YouTube, Facebook, Twitter, etc. A hateful message is defined as a statement that explicitly or implicitly expresses hatred or violence against people with protected characteristics (**S1 File**). This definition distinguishes hateful messages from offensive language by their targets: though offensive language can be directed at either individuals or groups (i.e.," Get out of this place!"), it does not target them based on their protected characteristics. Hateful messages are detrimental to both individuals and society because they can lead to prejudices against individuals, depreciation of minority's abilities, alienation of

**Data Availability Statement:** Publicly available datasets were analyzed in this study. This data can be found here: https://www.kaggle.com/parthplc/facebook-hateful-meme-dataset. The code of our model can be found here: https://github.com/SPQRXVIII/hateful_meme_detection. The codes of

baselines are summarized on the website here:
https://github.com/facebookresearch/mmf/tree/
main/projects/hateful_memes.

**Funding:** The authors received no specific funding
for this work.

**Competing interests:** The authors have declared
that no competing interests exist.

minorities, degradation of individual mental health, rise of suicide, increase in offline hate
crimes, and discriminatory practices when allocating public resources [1–6]. Especially due to
their massive scale and rapid propagation on the internet, hateful messages have become a crit-
ical threat to individuals and the whole society.

With tremendous social pressure to limit the spread of hateful messages, though no legal
framework explicitly targeting this subject has yet been developed, most major digital social
media platforms have developed their respective moderation policies [7,8]. However, the out-
dated manual moderation is considered insufficient, as human moderators are slow and
expensive. Moreover, content moderators are known to suffer post-traumatic stress disorder
(PTSD)-like syndromes after repetitively reviewing violent and exploitative content [9]. As a
result, generations of automatic hateful message detection methods have been developed since
2009, from the initial block-word-list approach to the current deep learning-based technology
[10–17]. Though earlier methods have been limited in their usefulness as they often can only
handle obvious patterns, encouragingly, the present transformer-based deep learning models
have reached a much higher accuracy (Acc.) of more than 90% in detecting textual hateful
messages due to its powerful capacity in the semantic comprehension [17]. The commercial
application of deep learning models effectively reduces the spreading of textual hateful
messages.

However, in recent years, a new type of "hateful message", namely "hateful meme", has
emerged and become rampant due to its expressiveness and subtlety [18]. Hateful memes
combine both text captions and background images to express users' intents. Specifically, hate-
ful memes can be more disguiseful than individual hateful images or hateful texts because
many sentences or images that are harmless or even hilarious may become hateful when placed
together (**Fig 1**). These subtle references are easy for humans to understand yet difficult for
machines to detect. So, hateful users thus intentionally publish hateful memes to evade con-
ventional detectors [18]. Traditional automatic text detection completely ignores image fea-
tures and vice versa. Therefore, the task of detecting multimodal hateful memes is very
challenging. In order to develop a detection model to capture the complexity of these memes,
the model should be not only able to process every single modality but also capable of fusing
the two modalities.

In the past few years, the advent of transformer-based large-scale Pre-trained Models
(PTMs) has shed hope into overcoming the challenge mentioned above. PTMs such as GPT
(Generative Pre-trained Transformers) and BERT (Bidirectional Encoder Representation
from Transformer) have recently achieved great success in many complex natural language
processing (NLP) tasks, demonstrating overall superior performance over conventional deep
learning models and becoming a milestone in the wider machine learning community [19].
The rich knowledge stored implicitly in the tremendous amount of model parameters could be
leveraged by fine-tuning them for specific downstream tasks. Following the success of PTMs
on language tasks, the deep-learning community has proposed various unified Vision-Lan-
guage PTMs (VL-PTMs), such as ConcatBERT, VilBERT, and Visual BERT, for multimodal
tasks [18,20–24]. These models are shown to be able to capture more complex information
and can be optimized to directly combine learned vector representations of both the text and
image to reach state-of-the-art results in several Vision-Language tasks [18,22,23,25,26]. How-
ever, no matter which method was used, the current VL-PTMs presented an Acc. of about
60% to 65% in the hateful meme detection task, which is still far away from human perfor-
mance (Acc. reported to be 84.7%) [18].

Although current VL-PTMs showed overwhelmingly positive performance in learning text
semantics, they do not fare so well when learning image region features, probably due to an
issue known as "ambiguity" [27]. To overcome this problem and improve the cross-modal

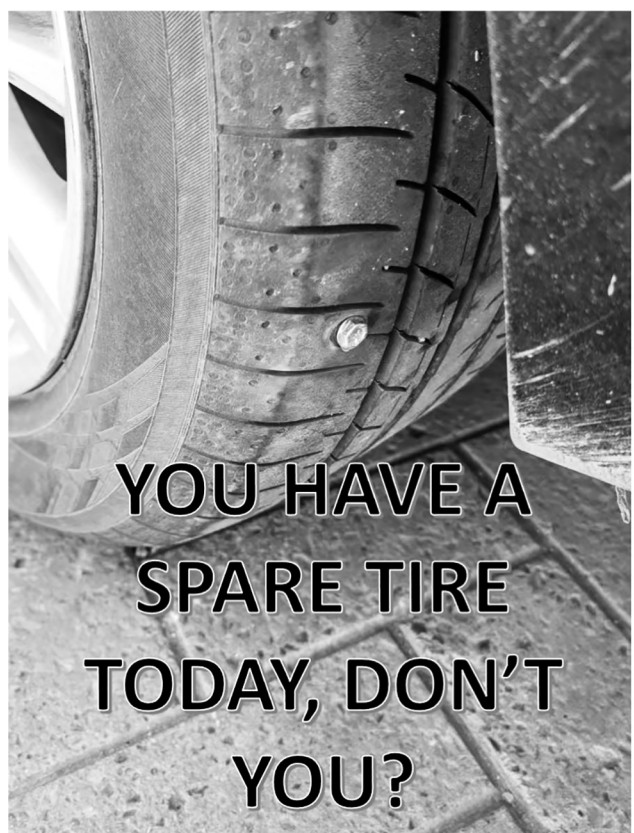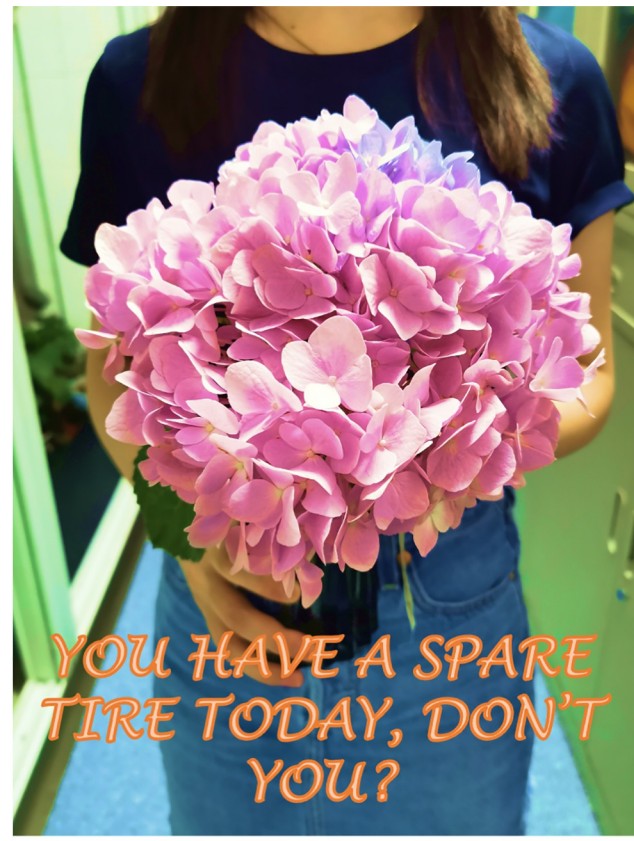

**A** **B**

**Fig 1. Examples of hateful and non-hateful memes.** In meme A (non-hateful), the image of a nailed tire and the sentence "You have a spare tire today, don't you?" are both neutral, as well as the combined version; in meme B (hateful), the same caption is inscribed as meme A, but the image part is changed from a nailed tire to the "girl holding flowers", has a reversed implication from neutrality to insulting as hateful. Besides, in meme B (hateful), neither the image part nor the caption expresses hatred alone, yet when combined, this meme expresses hatred. (Copyrighted by author F. Pan).

learning in Vision-Language semantic space, additional input of "anchor point" was added to our detection model to bridge the text and vision semantic spaces, which might improve the VL-PTM performance. This idea of "anchor point" is motivated by the observation that modern object detectors can accurately detect the salient objects in an image, which are often mentioned in the paired text data [24]. Utilizing anchor point information has been demonstrated to benefit cross-modal learning in VL-PTM training, as such models have shown an overall improvement over SoTA (state of the art results) in different downstream tasks, such as image retrieval, text retrieval, and image captioning [27]. However, this method has not been used in hateful meme detection yet. In this study, we built up a novel hateful meme detector by applying a transformer-based VL-PTM with a triplet input: image feature, text caption, and object tags identified in the image. Unlike previous VL-PTMs, our model additionally considers object tags of the image background, serving as the anchor points. We speculate that adding the object-tag input can bridge the text caption and related visual image in hateful meme learning and help facilitate visual feature classifications. After training and testing our model, we compare its performance against other published baselines in the same public dataset.

## Materials and methods

### Dataset

Our model was trained and tested in a public dataset named "Facebook Hateful Meme Dataset" (https://www.kaggle.com/parthplc/facebook-hateful-meme-dataset). In constructing this dataset, researchers at Facebook first reconstruct online memes by placing meme text caption over a new underlying licensed image without the loss of meaning. Then, they hired annotators from a third-party annotation company rather than a crowd-sourcing platform. The annotators were trained for 4 hours to recognize hateful memes according to Facebook guidelines (https://www.facebook.com/communitystandards/hatespeech) (**S1 File**). Then, these annotators reconstructed the memes and made the classification. For every hateful meme, there is always a non-hateful alternative whose caption or image is changed from the original one. This kind of substitution is called "benign confounders", a technique similar to current strategies of using counterfactual or contrastive examples; after the memes were labeled hateful, "benign confounders" were constructed [18]. Finally, a dataset with a total of 10k memes was set up and categorized into hateful or non-hateful. A dev and test set constituted 5% and 10% of the data, respectively, and the rest served as a training set.

### Detection pipeline

In this model, an optical character recognition (OCR) module is applied to extract the text part of the memes, and an object detection (OD) module VinVl (Visual features in Vision-Language) is used to encode the correlated image part of memes. Then, we choose a VL-PTM module named OSCAR+ (Object-SemantiCs Aligned Re-training) to encode the extracted text part, image part, and object tags of memes [24,28]. OSCAR+ was established on the basis of multi-layer Transformers like most other PTMs [29]. However, unlike most existing VL-PTMs only providing a dual input of concatenated image region features and text features, OSCAR + additionally affords a piece of specific information for object tags to bridge the text caption and related visual image.

We used the Google Colab platform to provide accelerators for inference and training. The hardware accelerators we used are an Nvidia P100 GPU and an 8-core TPU (Tensor 209 Processing Unit) V3. We first fine-tune OSCAR+ on the hateful meme dataset. In this stage, a two-layer Fully-Connected network (FC) is connected to OSCAR+ output (OSCAR+FC). The mini-batch gradient descent is carried out on the training set of 8500 images with a batch size set to 50 and a learning rate of 5.00E-06; the loss function is set to Binary Cross-Entropy Loss with Logits $L(x, y) = -(y \ln \sigma(x) + (1 - y) \ln (1 - \sigma(x))$. The dev set is used to optimize the model in avoidance of overfitting, and the final model is chosen at the epoch when the maximum area under the receiver operating characteristic (AUROC) is obtained on the dev set [30,31]. After OSCAR+ is fine-tuned, its output is connected to a random forest classifier (RF). Then, the RF is further optimized, consisting of ten decision trees whose maximum depth is set to 10. The trained RF is the final classifier (OSCAR+RF) for recognizing hateful memes. All training was repeated four times. The pipeline of our model construction is illustrated in **Fig 2**.

### Meme preprocessing

Because the meme itself can not be fed directly into OSCAR+, it must be preprocessed into a suitable data format (**Fig 3**). In preprocessing, the meme is first input into a VinVl object detector which uses the ResNeXt 152-C4 as the backbone feature extractor [32,33]. In VinVl, the backbone network transforms the input meme into a preliminary feature map. Afterwards, a Region Proposal Network (RPN) outlines regions of interest (ROIs) on the feature map

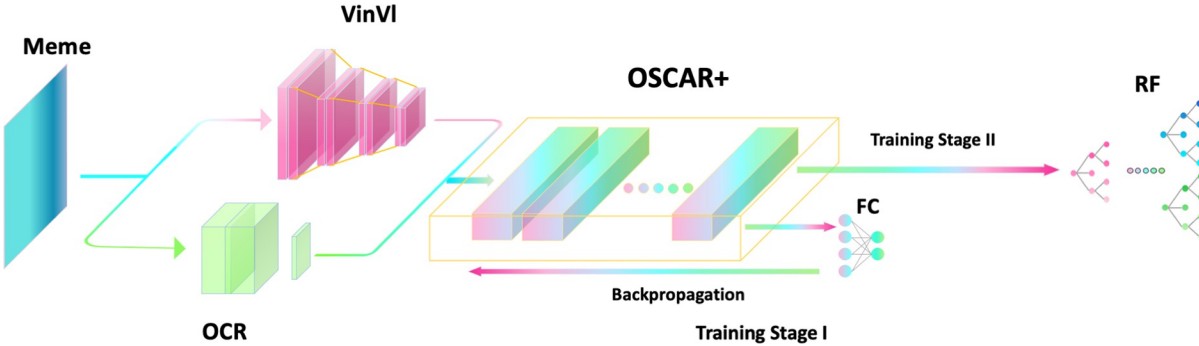

**Fig 2. Full pipeline.** FC denotes Fully-Connected network; RF denotes random forest classifier.

containing predefined categories of objects. The features of ROIs are then pooled into ROI-Pooling vectors of 2048 dimensions. Afterwards, those vectors are passed through two paralleled FCs pathways: 1. the first FC is used to predict the bounding box's position and size for each ROI-Pooling vector, outputting spatial features; then, each ROI-Pooling vector is

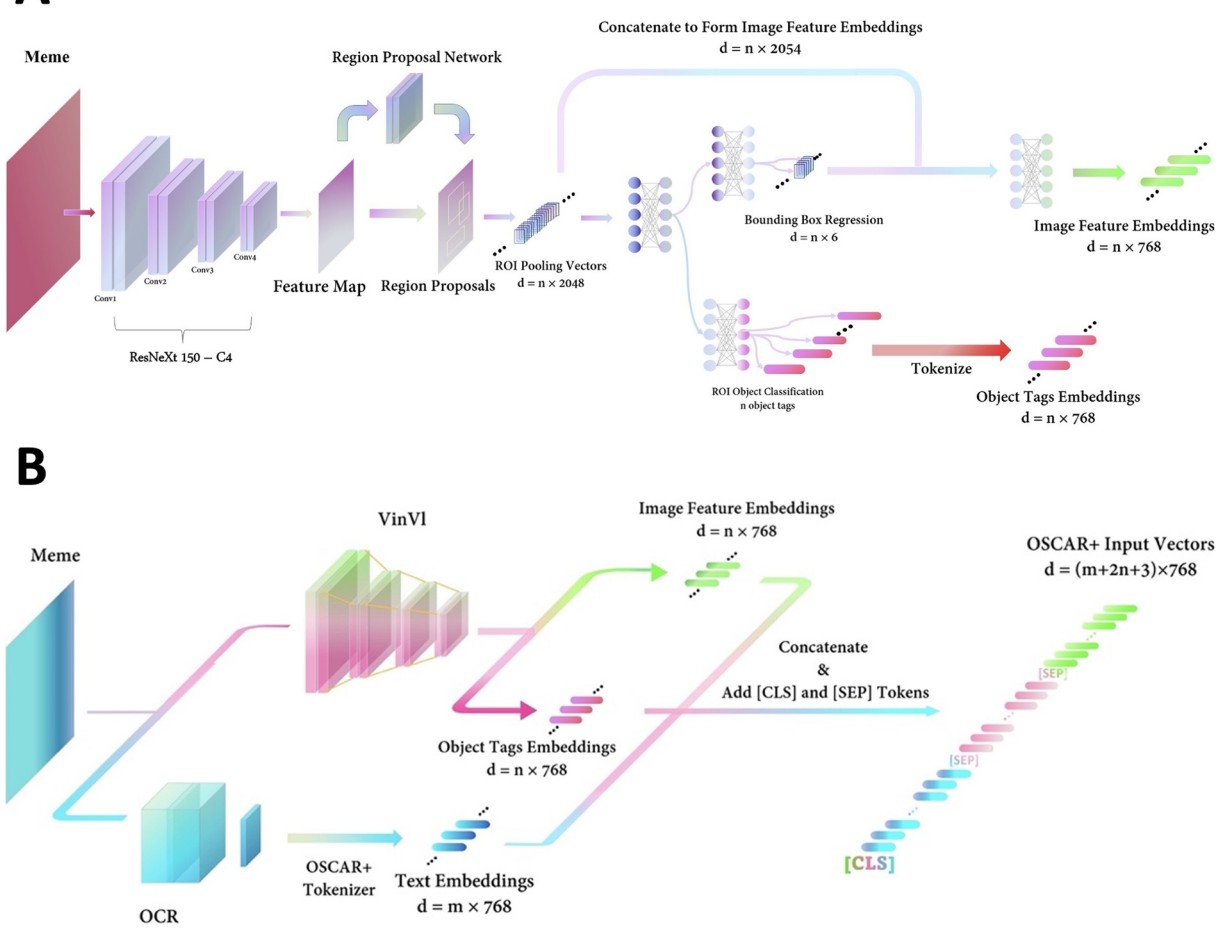

**Fig 3. Meme preprocessing.** A. Embeddings of image features and object tags in VinVl. B. Meme preprocessing by VinVl and OCR in parallel results in a triplet input of image feature, object tags, and text caption to OSCAR+.

concatenated with the corresponding Bounding-Box-Regression vector to pass through the following FC to form an image feature embedding vector with a size of 768; 2. the other paralleled FC is used to predict the category of the object in the corresponding ROIs outputting text embeddings of object tags with an exact size of 768. In general, the VinVl object detector will produce two sequences of vectors after meme input: the text object tags and the corresponding image feature embeddings (**Fig 3A**) [28].

On the other hand, the meme is input into an OCR module to extract all the caption text that appears in the memes [34]. Then, both text caption extracted by OCR and texted object tags outputted from VinVl were tokenized by OSCAR+, forming tokenized vectors with the same size of 768. At last, the text tokens, the object tag tokens, and the image feature embeddings are then further concatenated in sequence, inserting a special token [**SEP**] between different sections of OSCAR+ input and appending a special token [**CLS**] at the start of the whole input sequence. Afterwards, a meme was transferred into an embedded and tokenized triplet for further OSCAR+ encoding (**Fig 3B**) [24].

## Encoding process in VL-PTM

The OSCAR+ model consists of 12 same encoder blocks (**Fig 4A**) [24]. The first encoder takes the preprocessed meme embeddings with a matrix of dimension $(m + 2n + 3) \times 768$ as input, which consists of the image features embeddings ($m$), the object tag embeddings ($n$), the text caption embeddings ($n$) and the embedding of special tokens ([**CLS**], [**SEP**], and [**SEP**])**.** The followed encoder block takes the output embedding sequence produced by the previous encoder as input (**Fig 4A**). In every encoder block (**Fig 4B**), the input embedding sequence is passed through 12 self-attention heads in parallel, each outputting a smaller matrix of dimension $(m + 2n + 3) \times 64$. More specifically, in each self-attention head, the input matrix will pass through three paralleled FCs to produce three smaller matrices $Q$, $K$, and $V$ of dimension $(m + 2n + 3) \times 64$. Then, the standard dot product attention operation was carried out as:

$$Attention(Q, K, V) = softmax(\frac{QK^T}{\sqrt{d_k}})V, \qquad (1)$$

where $d_k$ equals 64 (**Fig 4C**) [29]. These intermediate output matrices are again concatenated to form a matrix of the original size, passed through an FC, then added with the original matrix, and normalized by rows. This normalized matrix is given through an FC, added with itself, and normalized by rows. Finally, the encoder block will produce a matrix of dimension $(m + 2n + 3) \times 768$.

## Comparisons with baselines

We compared the hateful meme detection Acc. and AUROC between our model and other published eleven (11) baselines trained on the same dataset [18]. The baselines included both unimodal PTMs and multimodal VL-PTMs. The unimodal PTMs pre-trained in text data was BERT (**Text BERT**) [14]. The unimodal PTMs pre-trained in image data included standard ResNet-152 with average pooling (**Image-Grid**) and with fine-tuned FC6 layer by using weights of the FC7 layer (**Image-Region**) [35]. The multimodal VL-PTMs include fused unimodal ResNet-152 and BERT output scores (**Late Fusion**) [18], concatenate ResNet-152 features with BERT and training an MLP on top (**Concat BERT**) [18], supervised multimodal bi-transformers using either Image-Grid or Image-Region feature input (**MMBT-Grid** and **MMBT-Region**) [36], ViLBERT [21], **Visual BERT** [37], ViLBERT trained on Conceptual Captions (**ViLBERT CC**) [38], and Visual BERT trained on COCO dataset (**Visual BERT COCO**) [39]. We re-evaluated these eleven models in our study on the same Google Colab

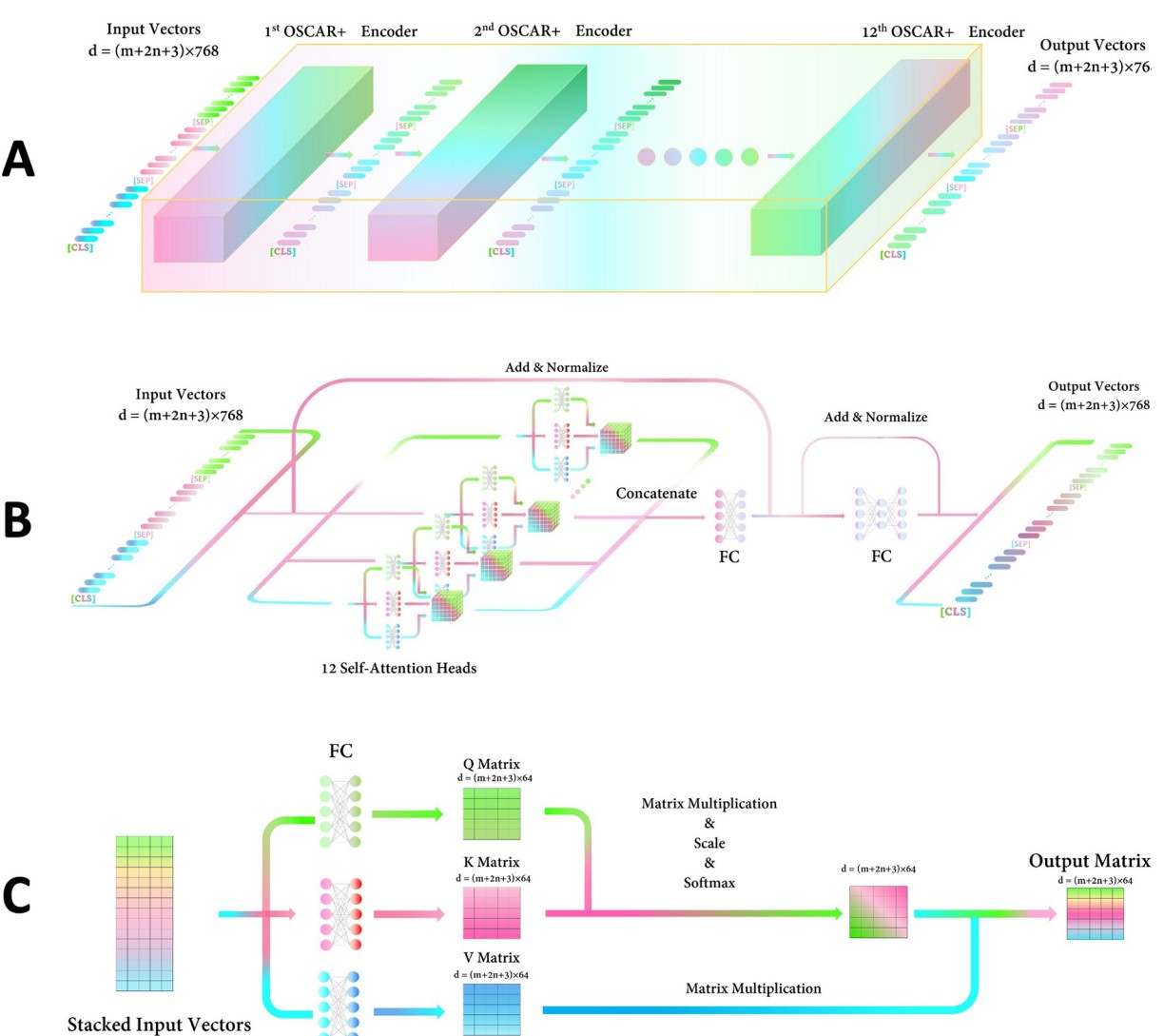

**Fig 4. Architectures of OSCAR+ and its encoder.** A. OSCAR+ consists of 12 tandem encoders; B. the architecture of an encoder; C. structure of the self-attention head.

platform. Batch sizes were chosen to meet memory constraints. As mentioned before, the training of all models was consecutively repeated four times to select four final models. Afterwards, all these final models, including ours and baselines, were evaluated in the test set, and average values of Acc. and AUROC were compared.

## Results

The results are summarized in **Table 1**. Our proposed OSCAR+RF model achieved the best performance at the test set among all models with an average Acc. of 0.684 and an average AUROC of 0.768, followed by OSCAR+FC with an average Acc. of 0.677 and an average AUROC of 0.762 with very slight differences. Besides, we observe that the text-only classifier performs slightly better than the vision-only classifier, and multimodal VL-PTMs performed better than the unimodal models. In multimodal detecting models, adding the tag information (OSCAR+RF and +FC) increases both the Acc. and AUROC.

**Table 1. Comparisons among our models with other published baselines.**

| Models | Batch size | Detecting modality | Loss | Optimizer | Learning rate | Acc., dev set (n = 500)* | AUROC, dev set (n = 500) * | Acc., test set (n = 1000) * | AUROC, test set (n = 1000) * |
|---|---|---|---|---|---|---|---|---|---|
| **Image-Grid** | 32 | Image | Cross entropy | AdamW | 1.00E-05 | 0.500±0.045 (0.436–0.536) | 0.516±0.027 (0.478–0.543) | 0.511±0.023 (0.478–0.526) | 0.514±0.018 (0.498–0.530) |
| **Image-Region** | 32 | Image | Cross entropy | AdamW | 5.00E-05 | 0.513±0.032 (0.484–0.548) | 0.549±0.030 (0.508–0.579) | 0.531±0.023 (0.502–0.558) | 0.561±0.039 (0.526–0.617) |
| **Text BERT** | 64 | Text | Cross entropy | AdamW | 5.00E-05 | 0.569±0.020 (0.548–0.588) | 0.625±0.047 (0.579–0.669) | 0.586±0.024 (0.556–0.612) | 0.639±0.006 (0.633–0.645) |
| **Late Fusion** | 32 | Image&Text | Cross entropy | AdamW | 5.00E-05 | 0.589±0.031 (0.544–0.612) | 0.641±0.040 (0.613–0.700) | 0.619±0.011 (0.608–0.630) | 0.679±0.018 (0.665–0.705) |
| **ConcatBERT** | 32 | Image&Text | Cross entropy | AdamW | 1.00E-05 | 0.576±0.038 (0.540–0.616) | 0.645±0.012 (0.629–0.655) | 0.622±0.023 (0.588–0.636) | 0.682±0.017 (0.659–0.696) |
| **MMBT-Grid** | 32 | Image&Text | Cross entropy | AdamW | 1.00E-05 | 0.603±0.042 (0.544–0.644) | 0.672±0.018 (0.654–0.696) | 0.631±0.014 (0.616–0.650) | 0.694±0.006 (0.687–0.700) |
| **MMBT-Region** | 32 | Image&Text | Cross entropy | AdamW | 5.00E-05 | 0.605±0.059 (0.524–0.652) | 0.649±0.067 (0.585–0.722) | 0.642±0.032 (0.608–0.672) | 0.690±0.046 (0.646–0.735) |
| **ViLBERT** | 32 | Image&Text | Cross entropy | AdamW | 1.00E-05 | 0.633±0.020 (0.612–0.656) | 0.717±0.035 (0.677–0.747) | 0.659±0.007 (0.652–0.668) | 0.732±0.015 (0.716–0.753) |
| **Visual BERT** | 32 | Image&Text | Cross entropy | AdamW | 5.00E-05 | 0.638±0.023 (0.612–0.668) | 0.722±0.010 (0.711–0.732) | 0.664±0.013 (0.656–0.684) | 0.748±0.011 (0.732–0.757) |
| **ViLBERT CC** | 32 | Image&Text | Cross entropy | AdamW | 1.00E-05 | 0.656±0.009 (0.648–0.668) | 0.730±0.035 (0.691–0.773) | 0.664±0.009 (0.652–0.674) | 0.739±0.016 (0.724–0.757) |
| **Visual BERT COCO** | 32 | Image&Text | Cross entropy | AdamW | 5.00E-05 | 0.648±0.032 (0.608–0.676) | 0.732±0.017 (0.711–0.752) | 0.664±0.020 (0.646–0.692) | 0.737±0.025 (0.711–0.770) |
| **OSCAR+FC** | 50 | Image&Tag&Text | Cross entropy | AdamW | 5.00E-06 | 0.666±0.038 (0.626–0.706) | 0.758±0.042 (0.703–0.803) | 0.677±0.010 (0.664–0.689) | 0.762±0.016 (0.749–0.786) |
| **OSCAR+RF** | 50 | Image&Tag&Text | Cross entropy | AdamW | 5.00E-06 | 0.667±0.034 (0.618–0.698) | 0.759±0.014 (0.745–0.777) | 0.684±0.002 (0.682–0.686) | 0.768±0.021 (0.737–0.784) |

**Footnotes:** Acc., accuracy; AUROC, area under the receiver operating characteristic.

*Mean±standard error with the range was calculated from evaluations of four final models.

## Discussion

In this study, we show that learning cross-modal representations can be improved by introducing object tags detected in memes as anchor points to ease the understanding of semantic alignments between images and text captions. Our results demonstrate that our multimodal VL-PTMs by intaking object tags are better than previous unimodal and multimodal models without anchor points in previous studies.

The detection accuracy of traditional unimodal deep learning methods, such as convolutional neural networks, was limited due to their consideration of only one modality; so, they could not comprehensively understand the subtlety behind the memes [10,16,18,19]. Our study shows that the unimodal models, including Image-Grid, Image-Region, and Text BERT, demonstrate lower Acc. and AUROC than any multimodal models. However, when comparing different unimodal models, the unimodal linguistic model (Text BERT) shows a better performance than visual unimodal models (Image-Grid and Image-Region) in detecting hateful memes, probably because the image region features are less representative of the meme's intention than word semantics [24]. In our results, Text BERT has an average AUROC of 0.639, higher than Image-Grid and Image-Region with only visual detecting modality (an average AUROC of 0.514 and 0.561, respectively).

On the other hand, multimodal learning of both text and image generally increases detection accuracy over unimodal learning. As mentioned before, many sentences or images that are harmless by themselves may become hateful when combined, or changing the image part or text part in the meme could quickly reverse the hateful intention. Thus, enhancing the cross-modal unification ability of the models can effectively increase the meme classification accuracy [20–24,37]. In addition, injecting more human knowledge into the model might help identify hateful memes more accurately. Encouragingly, the advent of large-scale PTM offered promising directions. Thanks to the immensity of training data and the massive number of model parameters (e.g., for BERT, the pre-training corpus contains 3,300 million words and the Base-version module has 110 million parameters while the Large-version module contains 340 million parameters), transformer-based VL-PTMs demonstrated higher potentials in complex learning because of the rich knowledge infused in the massive amount of model parameters; moreover, some have even surpassed human performance on multiple language understanding benchmarks, such as GLUE [19,35,40–42]. In a previous study, VL-PTMs were shown to be capable of re-learning from additional knowledge datasets, such as the Conceptual Captions dataset learned by ViLBERT CC and COCO dataset learned by Visual BERT COCO, presenting a potential benefit in the hateful meme detection when compared with the same baselines (ViLBERT and Visual BERT) without being trained those datasets [18]. But our study didn't find apparent discrepancies between these models. We believe pre-training these models with the more comprehensive caption or image entity datasets in future probably will improve the hateful meme detection accuracy because most hateful memes employ a lot of background knowledge which makes the relations between visual and text elements very complex and diverse [25,43,44].

More importantly, our results revealed that by intaking object tags as anchor points, our VL-PTM can achieve better Acc. and AUROC than conventional convolutional neural networks and previous VL-PTMs. This benefit is probably due to the more explicit representation by object tags which provide the model with clues about the corresponding image and caption features. Our model uses an OD module VinVl to encode and output a diverse tag collection of visual objects [28]. Promisingly, more powerful OD modules and web entity recognition Application Programming interfaces (API) have been developed and released over time. Combining these state-to-the-art tools probably facilitates more accurate object tag formation and, as a result, enhances the Vision-Language semantic learning by the VL-PTMs.

At last, we tried to replace the final FC linear classifier at the end of OSCAR+ with an RF classifier. After fine-tuning, the OSCAR+RF model shows a slight improvement in Acc. and AUROC compared with the OSCAR+FC model on the same test set. However, this finding somewhat supports the speculation that the merits of our model are majorly attributed to the addition of object tags input. The memes that invoked the combined visual and textual cues could be classified more correctly after strengthening the cross-modal association. Moreover, we conjecture that utilizing other more specific object tags, such as ethnic groups, nationalities, and religions, can provide further merits, posing a promising future research direction.

Our detection model also has some limitations. First, hateful messages are evolving quickly, so the model cannot keep its detection accuracy if not re-trained in time. For example, modern online communication heavily employs non-standard features, such as emojis and other irregular tokens such as $; and hateful users often try to evade detection by substituting the characters in their messages with symbols very different in terms of machine encodings yet looking or sounding very similar to human beings. One future research direction is to take advantage of these underutilized visual or audio aspects of the text information in order to adapt to more real-life scenarios. Second, the detection accuracy of our models, or similar models, largely depends on the knowledge base that the VL-PTM trained on. As we showed that detecting

underlying hateful metaphors requires the system to possess the ability to relate visual and linguistic entities in the image or captions to very detailed real-world knowledge, we expect future VL-PTMs to be supplemented with fine-grained external knowledge bases, such as Dbpedia [45] and wikidata [46] to achieve better performance on this task. Third, a limited capacity for object recognition in our OD module. So far, most contemporary VL-PTMs only take image and text into account, and the training dataset contains only common objects. However, the hateful memes often invoke unusual and specific objects connected to historical or social events not presented in those training sets. Thus, a sufficient dataset related to social or historical events is also in great demand to train a more knowledgeable OD module.

## Conclusion

This study has demonstrated that VL-PTMs with the addition of anchor points can improve the detection of hateful memes that involve strongly correlated text and visual information; our proposed models show a better detection performance than previous unimodal and multimodal baselines without anchor point applications.

## Supporting information

**S1 File. Definitions of hateful message (or hate speech).**
(DOCX)

## Acknowledgments

I would like to express my sincere gratitude to Prof. Xingang Wang at School of Electronic Information and Communications (EIC), Huazhong University of Science and Technology for many professional suggestions on the artificial intelligence model establishment.

## Author Contributions

**Conceptualization:** Yuyang Chen, Feng Pan.

**Data curation:** Yuyang Chen, Feng Pan.

**Formal analysis:** Yuyang Chen, Feng Pan.

**Investigation:** Yuyang Chen, Feng Pan.

**Methodology:** Yuyang Chen, Feng Pan.

**Project administration:** Yuyang Chen, Feng Pan.

**Resources:** Yuyang Chen, Feng Pan.

**Software:** Yuyang Chen, Feng Pan.

**Supervision:** Feng Pan.

**Validation:** Yuyang Chen, Feng Pan.

**Writing – original draft:** Yuyang Chen.

**Writing – review & editing:** Feng Pan.

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
