## [Decision Letter · Decision Letter 0]

14 Jun 2022

PONE-D-22-10554Multimodal Detection of Hateful Memes by Applying a Vision-Language Pre-Training Model PLOS ONE

Dear Dr. Pan,

Thank you for submitting your manuscript to PLOS ONE. After careful consideration, we feel that it has merit but does not fully meet PLOS ONE’s publication criteria as it currently stands. Therefore, we invite you to submit a revised version of the manuscript that addresses the points raised during the review process.

We look forward to receiving your revised manuscript.

Kind regards,

Liejun Wang, Ph.D.

Academic Editor

PLOS ONE

Journal Requirements:

2. We note that S1 Fig and S2 includes an image of a participant in the study. 

Reviewers' comments:

Reviewer's Responses to Questions

**Comments to the Author**

1. Is the manuscript technically sound, and do the data support the conclusions?

Reviewer #1: Yes

Reviewer #2: Yes

2. Has the statistical analysis been performed appropriately and rigorously? 

Reviewer #1: Yes

Reviewer #2: Yes

3. Have the authors made all data underlying the findings in their manuscript fully available?

Reviewer #1: Yes

Reviewer #2: Yes

4. Is the manuscript presented in an intelligible fashion and written in standard English?

Reviewer #1: Yes

Reviewer #2: Yes

5. Review Comments to the Author

Reviewer #1: This study has demonstrated that Vision-Language PTMs with the addition of anchor points can improve the detection of hateful memes that involve strongly correlated textual and visual information. However, I have the following issues:

1. The motivation of the paper is not clear in Section Introduction. What are the challenges involved? What solutions already exist for the problem you want to solve? What are their limitations and drawbacks? It lacks a critical analysis of these works and a positioning of the proposed solution concerning the state-of-the-art works.

2. It is not clear the contribution of this work for the advance of the state-of-the-art in this domain.

3. The experimental results may be discussed in detail fundamentally by providing critical analysis.

4. Add most recent references at appropriate places. Such as:

https://link.springer.com/article/10.1007/s00521-021-06546-x

https://link.springer.com/article/10.1007/s00521-022-07054-2

Reviewer #2: This paper presents a mode for hateful memes detection by integrating vision-language pre-trained models with random forest classifier. Experiments are conducted on Facebook Hateful Meme Dataset to evaluate the proposed model.

Although the main idea is easy to follow, I have several concerns as following:

- The main idea of this work seems a simple combination of two existing works (OSCAR and random forest classifier).

- Several related works for vision-language pre-training are missing:

[A] "Unified Vision-Language Pre-Training for Image Captioning and VQA." AAAI. 2020.

[B] “Scheduled Sampling in Vision-Language Pretraining with Decoupled Encoder-Decoder Network.” AAAI. 2021.

- In Table 1, the performance in Test Acc. of OSCAR+RF is lower than Visual BERT COCO.

- More ablated runs (e.g., Visual BERT COCO+RF) should be included for performance comparison.

6. PLOS authors have the option to publish the peer review history of their article (what does this mean?). If published, this will include your full peer review and any attached files.

Reviewer #1: No

Reviewer #2: No

---

## [Author Response · Author response to Decision Letter 0]

16 Jul 2022

Response to Reviewers

Editors:

Journal Requirements:

2. We note that S1 Fig and S2 includes an image of a participant in the study. 

R: We appreciate your notifications on the manuscript style and informed consent of the cited figures. In the modification, we revised our manuscript following your comments and added the agreement of the figure publish.

Reviewer #1:

This study has demonstrated that Vision-Language PTMs with the addition of anchor points can improve the detection of hateful memes that involve strongly correlated textual and visual information. However, I have the following issues:

1. The motivation of the paper is not clear in Section Introduction. What are the challenges involved? What solutions already exist for the problem you want to solve? What are their limitations and drawbacks? It lacks a critical analysis of these works and a positioning of the proposed solution concerning the state-of-the-art works.

2. It is not clear the contribution of this work for the advance of the state-of-the-art in this domain.

3. The experimental results may be discussed in detail fundamentally by providing critical analysis.

4. Add most recent references at appropriate places. Such as:

https://link.springer.com/article/10.1007/s00521-021-06546-x

https://link.springer.com/article/10.1007/s00521-022-07054-2

R: We appreciate your efforts to point out these flaws in the manuscript, and we followed your suggestions to revise our draft. First, in the introduction, we explained the importance and dilemma of hateful meme detection nowadays, including limitations and drawbacks of different techniques. Besides, we highlighted our innovation of adding tags as anchor points which might enhance the capability of hateful meme detection. 

Second, the idea of the anchor point in VLP is one of the most critical progresses in the past two years. When we started this project, this idea had not been used in the downstream task of hateful meme detection. We believe our findings could inspire fellow researchers to adopt and improve this thought. At the same time, in the discussion section, we also pointed out some future research directions, such as adding more specific tags, including race, skin color, etc., which we believe can improve the efficacy of the detector. 

Third, in the last few weeks, we consecutively retrained our model and other eleven baselines four times on the Google Colab platform for a comprehensive comparison. The batch size was set to the largest value given the memory restriction. Then, to reduce the variance of estimating the performance of each model, all of them were trained and tested four times, and the averages of Acc. and AUROC were taken for comparison. We observed that our multimodal model with anchor points performed better than unimodal or other multimodal models without anchor points. These results support our idea mentioned above. 

At last, we appreciate your recommending those two related papers. In revision, we cited them to support our method part. 

Reviewer #2:

This paper presents a mode for hateful memes detection by integrating vision-language pre-trained models with random forest classifier. Experiments are conducted on Facebook Hateful Meme Dataset to evaluate the proposed model.

Although the main idea is easy to follow, I have several concerns as following:

- The main idea of this work seems a simple combination of two existing works (OSCAR and random forest classifier).

- Several related works for vision-language pre-training are missing:

[A] "Unified Vision-Language Pre-Training for Image Captioning and VQA." AAAI. 2020.

[B] "Scheduled Sampling in Vision-Language Pretraining with Decoupled Encoder-Decoder Network." AAAI. 2021.

- In Table 1, the performance in Test Acc. of OSCAR+RF is lower than Visual BERT COCO.

- More ablated runs (e.g., Visual BERT COCO+RF) should be included for performance comparison.

R: We'd like to express our sincere gratitude for your review and for providing these valuable comments and suggestions. We revised our manuscript thoroughly and added some essential information about our model. First, our model is based on anOSCAR+ backbone, which differs from other vision-language transformers by taking images, anchor point tags, and text caption triplets as inputs. Although our model can be considered as a combination of an optical character recognition (OCR) module, an object detection (OD) module, a fine-tuned OSCAR+ module, and a random forest (RF) classifier module, this idea and our results are inspiring. Still, there is a substantial space for future improvement on the automatic hateful meme detection using deep-learning techniques. The emergence of multimodal transformer shed light on achieving effective detection of hateful memes owing to its capability in cross-modal context semantic learning. From previous studies, multimodal transformers presented superior performances (Acc. about 60-65%) compared to conventional CNN models (Acc. about 50% - 51%) in this task. However, they are still very far away from human performances (Acc. about 85%). Under this milieu, we noticed that cross-modal correlation (so-called "ambiguity" issue) might limit the performance of multimodal transformers. As a solution, anchor points, one of the most critical ideas in multimodal deep learning in the past two years, is adopted in this study to connect image and text semantics. When we started our project, this model type had not been attempted in meme classification yet. We believe the results demonstrated by our study would provide justification for future research on such methods. In future research, we also plan to go further by employing other techniques, such as adding more specific tags, including race, skin tone, etc. These upgraded tags have the potential to improve model performance. In our revision, we explained this issue in the introduction and discussion. 

Second, we appreciate you for providing us with those two papers about unified VL transformers with different designs. In the revision, we discussed applying different VL transformers in hateful meme detection. 

Third, about the question that the Acc. of OSCAR+RF in the test set is lower than Visual BERT COCO, we admit that the data in Table 1 is not rigorous. In our previous version, we simply cited the results from the published literature instead of running these models on our own. In this revision, we retrained those baseline models and added another two multimodal models (Late fusion and ConcatBERT) on the same Goggle Colab platform with batch sizes set to the largest possible value given memory limitation. In addition, all were consecutively trained and tested four times and the mean values of their Acc. and AUROC were compared, which helped to reduce the variance of estimating model performance. As a result, we can see Transformer-based Vision-Language Pre-Training Models (VL-PTMs) with anchor point tags inputs performed better than conventional VL PTMs without those tags. However, after multiple training, we find the differences of Acc. and AUROC between OSCAR+FC and OSCAR+RF are very small (actually, in previous manuscript, this gap was also small). So, we speculated that the benefit of our model is majorly ascribed to the addition of the anchor point involvement (object tags) but not the last classifier. Following your comments, we also revised our viewpoint and discussion.

---

## [Decision Letter · Decision Letter 1]

26 Aug 2022

Multimodal Detection of Hateful Memes by Applying a Vision-Language Pre-Training Model

PONE-D-22-10554R1

Dear Dr. Pan,

We’re pleased to inform you that your manuscript has been judged scientifically suitable for publication and will be formally accepted for publication once it meets all outstanding technical requirements.

Kind regards,

Liejun Wang, Ph.D.

Academic Editor

PLOS ONE

Additional Editor Comments (optional):

The reviewer has completed reviewing the manuscript and the authors have addressed the reviewer's concerns. Therefore, I recommend publishing the paper.

Reviewers' comments:

Reviewer's Responses to Questions

**Comments to the Author**

1. If the authors have adequately addressed your comments raised in a previous round of review and you feel that this manuscript is now acceptable for publication, you may indicate that here to bypass the “Comments to the Author” section, enter your conflict of interest statement in the “Confidential to Editor” section, and submit your "Accept" recommendation.

Reviewer #1: All comments have been addressed

Reviewer #2: All comments have been addressed

2. Is the manuscript technically sound, and do the data support the conclusions?

Reviewer #1: Yes

Reviewer #2: (No Response)

3. Has the statistical analysis been performed appropriately and rigorously? 

Reviewer #1: Yes

Reviewer #2: (No Response)

4. Have the authors made all data underlying the findings in their manuscript fully available?

Reviewer #1: No

Reviewer #2: (No Response)

5. Is the manuscript presented in an intelligible fashion and written in standard English?

Reviewer #1: Yes

Reviewer #2: (No Response)

6. Review Comments to the Author

Reviewer #1: Thank you for the response and revisions. You have successfully addressed the issues raised by the reviewers and the manuscript can be accepted for publication.

Reviewer #2: (No Response)

7. PLOS authors have the option to publish the peer review history of their article (what does this mean?). If published, this will include your full peer review and any attached files.

Reviewer #1: No

Reviewer #2: No

---

## [Editor Report · Acceptance letter]

2 Sep 2022

PONE-D-22-10554R1 

Multimodal Detection of Hateful Memes by Applying a Vision-Language Pre-Training Model 

Dear Dr. Pan:

I'm pleased to inform you that your manuscript has been deemed suitable for publication in PLOS ONE. Congratulations! Your manuscript is now with our production department. 

Kind regards, 

on behalf of

Dr. Liejun Wang 

Academic Editor

PLOS ONE